# Evaluation of kidney function among people living with HIV initiating antiretroviral therapy in Zambia

Jake M. Pry[1]*, Michael J. Vinikoor[2], Carolyn Bolton Moore[1,2], Monika Roy[3], Aaloke Mody[4], Izukanji Sikazwe[1], Anjali Sharma[1], Belinda Chihota[1], Miquel Duran-Frigola[5], Harriet Daultrey[6], Jacob Mutale[1], Andrew D. Kerkhoff[3], Elvin H. Geng[4], Brad H. Pollock[6], Jaime H. Vera[7]

1 Centre for Infectious Disease Research Zambia (CIDRZ), Lusaka, Zambia, 2 School of Medicine University of Alabama, Birmingham, Alabama, United States of America, 3 School of Medicine, University of California, San Francisco, California, United States of America, 4 School of Medicine, Washington University, St. Louis, Missouri, United States of America, 5 Ersilia Open Source Initiative, Cambridge, United Kingdom, 6 School of Medicine, University of California, Davis, California, United States of America, 7 Department of Global Health and Infection, Brighton and Sussex Medical School, University of Sussex, Brighton, United Kingdom

* jmpry@ucdavis.edu, jake.pry@cidrz.org

**Data Availability Statement:** The Government of Zambia allows data sharing after a review of data queries ensures the appropriateness of its intended use. To request data access, contact the CIDRZ

## Abstract

As the response to the HIV epidemic in sub-Saharan Africa continues to mature, a growing number of people living with HIV (PLHIV) are aging and risk for non-communicable diseases increases. Routine laboratory tests of serum creatinine have been conducted to assess HIV treatment (ART) suitability. Here we utilize those measures to assess kidney function impairment among those initiating ART. Identification of non-communicable disease (NCD) risks among those in HIV care creates opportunity to improve public health through care referral and/or NCD/HIV care integration. We estimated glomerular filtration rates (eGFR) using routinely collected serum creatinine measures among a cohort of PLHIV with an HIV care visit at one of 113 Centre for Infectious Disease Research Zambia (CIDRZ) supported sites between January 1, 2011 and December 31, 2017, across seven of the ten provinces in Zambia. We used mixed-effect Poisson regression to assess predictors of eGFR <60ml/min/1.73m$^2$ allowing random effects at the individual and facility level. Additionally, we assessed agreement between four eGFR formulae with unadjusted CKD-EPI as a standard using Scott/Fleiss method across five categories of kidney function. A total of 72,933 observations among 68,534 individuals met the inclusion criteria for analysis. Of the 68,534, the majority were female 41,042 (59.8%), the median age was 34 (interquartile range [IQR]: 28–40), and median CD4 cell count was 292 (IQR: 162–435). The proportion of individuals with an eGFR <60ml/min/1.73m$^2$ was 6.9% (95% CI: 6.7–7.1%) according to the unadjusted CKD-EPI equation. There was variation in agreement across eGFR formulas considered compared to unadjusted CKD-EPI ($\chi^2$ p-value <0.001). Estimated GFR less than 60ml/min/1.73m2, per the unadjusted CKD-EPI equation, was significantly associated with age, sex, body mass index, and blood pressure. Using routine serum creatinine measures, we identified a significant proportion of individuals with eGFR indicating moderate or great kidney function impairment among PLHIV initiating ART in Zambia. It is possible that differentiated

Ethics and Compliance Committee Chair/Chief Scientific Officer, Dr. Roma Chilengi, Roma. chilengi@cidrz.org, or the Secretary to the Committee/Head of Research Operations, Ms. Hope Mwanyungwi, Hope.Mwanyungwi@cidrz.org, mentioning the intended use for the data.

**Funding:** Funding for this work is provided by the President's Emergency Plan for AIDS Relief (PEPFAR) and Centers for Infectious Disease Research Zambia through a grant awarded to IS (Grant NU2GGH001920). The funders had no role in study design, data collection and analysis, decision to publish, or preparation of the manuscript.

**Competing interests:** None of the authors have competing interests.

service delivery models could be developed to address this subset of those in HIV care with increased risk of chronic kidney disease.

## Background

As the response to the HIV epidemic in sub-Saharan Africa continues to mature, a growing number of people living with HIV (PLHIV) are aging and there is growing risk for non-communicable diseases (NCD) such as chronic kidney disease (CKD) [1–5]. Many resource-limited countries in sub-Saharan Africa (SSA), including Zambia, follow the World Health Organization Guidelines to assess renal function prior to or at the time of starting antiretroviral therapy (ART) containing tenofovir disoproxil fumarate (TDF), which has potential for nephrotoxicity [6,7]. Routine measurement of serum creatinine also creates an opportunity for epidemiological analysis of kidney function impairment including acute kidney injury and chronic kidney disease. Presently, relatively few resources are allocated and available for follow-up assessment and referral in cases of high serum creatinine measures [8]. Better estimates of kidney function impairment can provide evidence and motivating rationale to expand NCD care guidance at ART care facilities and for HIV/NCD care integration to improve outcomes among those in HIV care in Zambia.

Chronic kidney disease prevalence is estimated to be 10.0% in Zambia [9]. Evidence among those in HIV care suggests that kidney function measures, such as creatinine clearance, are associated with progression to chronic kidney disease and early mortality globally [10–12]. Tenofovir disoproxil fumarate, and less so tenofovir alafenamide, have been linked with proximal tubular dysfunction, Fanconi syndrome, and acute kidney injury [13–16]. An assessment of kidney function impairment was conducted in 2019 at a single high-volume HIV care facility urban Zambia among ART-naïve individuals entering HIV care in 2011–2013; it found that 4.1% had moderate or severe eGFR measures (59–15 ml/min/1.73m$^2$) [13]. Despite routine collection of serum creatinine measures for HIV care, robust, generalizable analyses of large HIV cohorts regarding estimated glomerular filtration rate (eGFR) are scant. Furthermore, a gap remains in understanding predictors of kidney function impairment among those initiating ART in Zambia and sub-Saharan Africa.

We calculated eGFR from routine, programmatic HIV care measures recorded in the national electronic HIV medical record. Leveraging information in the medical record we assessed predictors of kidney function impairment (<60ml/min/1.73m2) using regression analysis. We also evaluated correlation between TDF-containing regimens and eGFR and compared six different formulae for eGFR. These findings can guide policy, care recommendations, and represent the opportunity for kidney/HIV care integration to improve health outcomes through spotlighting this high-risk group at the national level [17–20].

## Methods

### Design

We conducted a cross-sectional analysis on the outcome of estimated glomerular filtration rate among individuals initiating ART in Zambia using the national electronic HIV medical record. Additionally, among those with multiple measures in the HIV medical record we conducted descriptive analysis to identify trends in eGFR during a two years of follow-up period.

## Population

All individuals with an HIV care clinic visit recorded at one of the 113 HIV care sites supported by the Centre for Infectious Disease Research Zambia (CIDRZ) between January 1, 2011 and December 31, 2017, aged >16 to 80 years with at least one serum creatinine measure on record were eligible to be included in the analysis. HIV care data from all health facilities within seven of the ten Zambian provinces are recorded in the EMR; this includes both urban and rural settings as well as all levels of care (e.g., clinics and hospitals).

## Setting

The Centre for Infectious Disease Research Zambia (CIDRZ) is a non-governmental organization with a national scope conducting research and providing support in the form of public health, laboratory, and research training, program guidance and development through robust monitoring and evaluation, especially HIV and tuberculosis. CIDRZ maintains close partnerships with the Zambia Ministry of Health (MoH), and the Centers for Disease Control and Prevention Zambia to support HIV prevention, care, and treatment services, including mobile laboratory services to improve reach and coverage of critical laboratory services, in public clinics in both rural and urban settings across four of ten Zambian provinces, funding primarily through PEPFAR/CDC [21].

## Measures

Individuals receiving HIV care in Zambia are assigned a unique identifying number and undergo an initial history and physical examination and baseline laboratory tests. All data collected at an HIV care visit, including demographic, laboratory, and clinical information are recorded in the Zambian national HIV electronic medical record, SmartCare. Prior to ART initiation, the MoH recommends measurement of serum creatinine, as first-line ART regimens often include TDF, which has potential nephrotoxicity [22,23]. While specific recommendations for serum creatinine measures are not outlined beyond ART initiation, additional, follow-up measures may be ordered on a clinical, ad hoc, basis as well. Guidelines for ART initiation varied by required CD4 cell count and/or World Health Organization (WHO) stage during the analysis period [22–24]. All clinical data including serum creatinine results from initiation and subsequent clinical visits are recorded in the EMR [23]. Other covariates of interest that we used in this analysis included height, weight, date of birth, sex, time in care, previous diabetes diagnosis, ART regimen, pregnancy status, and blood pressure, which were abstracted from the initial history and physical, clinical follow-up, and/or short visit forms. Though the framework for capturing covariates exists in the electronic HIV medical record it is important to note that they are not required by the system.

Estimated glomerular filtration rates (eGFR) were categorized into one of five mutually exclusive categories in accordance with the United States National Kidney Foundation: normal kidney function (≥90 ml/min/1.73m$^2$), mild kidney function impairment (60-89ml/min/1.73m$^2$), moderate kidney function impairment (30-59ml/min/1.73m$^2$), severe kidney function impairment (15–29 ml/min/1.73m$^2$), and kidney failure (<15 ml/min/1.73m$^2$ [25]. Several formulae for eGFR were implemented including (1) CKD-EPI equation (adjusted and unadjusted for race), (2) Cockcroft-Galt equation (CG), (3) Mayo Quadratic Equation (Mayo), and (4) four-variable modification of diet in renal disease equation (adjusted and unadjusted for race) (MDRD-4) [26–30].

Estimated glomerular filtration rate (eGFR) calculations are as follows:

- $eGFR_{female}(CKD\ EPI) = 141 * min\left(\frac{SCr}{0.7}\right)^{-0.329} * max\left(\frac{SCr}{0.7}\right)^{-1.209} * 0.993^{Age} * 1.018 * (1.159\ if\ black)$

- $eGFR_{male}(CKD\ EPI) = 141 * min\left(\frac{SCr}{0.9}\right)^{-0.411} * max\left(\frac{SCr}{0.9}\right)^{-1.209} * 0.993^{Age} * (1.159\ if\ black)$

- $eGFR_{female}(CG) = [(140 - age) * weight * 0.85]/(72 * SCr)$

- $eGFR_{male}(CG) = [(140 - age) * weight]/(72 * SCr)$

- $eGFR_{female}(MDRD - 4) = 186 * (Serum\ Creatinine)^{-1.154} * Age^{-0.203} * (0.742) * (1.212\ if\ black)$

- $eGFR_{male}(MDRD - 4) = 186 * (Serum\ Creatinine)^{-1.154} * Age^{-0.203} * (1.212\ if\ black)$

- $eGFR(Mayo) = e^{\left[\left(1.911 + \frac{+5.249}{SCr}\right) - \left(\frac{2.114}{SCr_2}\right) - 0.00686 * Age - 0.205(if\ female)\right]}$

Note: All equations use mg/dL measures for serum creatinine (SCr) and age in years.

Blood pressure categorization was done in accordance with American Heart Association (AHA)/American College of Cardiology (ACC) 2017 guidelines except for severe hypertension defined as systolic pressure ≥180mmHg or diastolic pressure ≥120mmHg [31–34].

Body mass index (BMI) was calculated from weight (kg) and height (m) data recorded in the initial history and physical form extracted from the EMR where the following equation was applied:

$$body\ mass\ index = \frac{weight(kg)}{height^2(m)}$$

Observations of BMI outside the 6–50 range, were dropped from analysis. Multiple imputation was considered where missingness was <30%. Categories for BMI are defined according to World Health Organization criteria [35].

## Analysis

Descriptive statistics were compared using independent t-tests for continuous comparisons and contingency table analysis with $\chi^2$ tests for categorical comparisons. We conducted mixed-effects Poisson regression on eGFR <60ml/min per unadjusted CKD-EPI formula at baseline (ART initiation) without adjustment for race/ethnicity allowing random effects at facility level. For those with multiple creatinine measures we assessed the change in eGFR at baseline, three to twelve months and greater than twelve months. Unadjusted formulas were compared using Scott/Fleiss Pi agreement estimation with CKD-EPI as the comparison/referent formula. All analyses were completed using Stata 15 SE (StataCorp LLC, College Station, Texas USA) and figures were created using R Software 4.0.3 (R Foundation for Statistical Computing, Vienna, Austria).

## Ethical statement

The review of existing, de-identified, routinely collected programmatic data was approved by the U.S. Centers for Disease Control & Prevention (2018–381), University of Zambia Biomedical Research Ethics Committee (011-12- 17), University of North Carolina at Chapel Hill, USA (18–0854) and the Institutional Review Board at Washington University, St. Louis, USA (2019–11143).

## Results

A total of 467,178 individuals were recorded in the national electronic HIV medical record from January 1, 2011 through December 31, 2017 among which 68,628 (14.7%) met the inclusion criteria and a total of 72,933 unique observations (3,209 individuals had multiple measures on record) were included in the analysis dataset. Of the 68, 628, the majority were

women (59.8%), the median age was 34 years (interquartile range [IQR]: 28–40 years), and the median CD4 cell count was 292 (IQR: 162–435) (**Table 1**). The median body mass index was within normal weight category limits at 20.7 (IQR: 18.5–23.4) with 16.6% categorized as over-weight or obese.

## Prevalence of kidney function impairment

We found that 6.9% (95% CI: 6.7–7.1%) of those reviewed had an eGFR $<60$ml/min/$1.73$m$^2$, per CKD-EPI formula (**Table 1**). The median eGFR (CKD-EPI) among men was higher $101.9$ml/min/$1.73$m$^2$ (IQR: 83.9–115.5ml/min/$1.73$m$^2$) compared to women (96.3ml/min/$1.73$m$^2$ IQR: 79.7–114.3ml/min/$1.73$m$^2$). Severe kidney function impairment or kidney failure were observed among 2.5% (95% CI: 2.1, 2.9%) of those 50 years of age and older (**Fig 1A**). The crude prevalence ratio was highest for those in the 65+ age group at 11.22 (95% CI: 9.40, 13.40) compared to those aged 17–24 years (**Table 2**).

## Kidney function impairment and blood pressure

The median systolic and diastolic blood pressure among 39,566 (57.3%) for which a blood pressure measure was recorded was 110mmHg (IQR: 100-120mmHg) and 70mmHg (IQR: 60-80mmHg), respectively (**Table 1**). We observed an inverse relationship between eGFR and blood pressure, (**Fig 1B**). The crude prevalence ratio was highest for those in the severe hypertension category at 2.81 (95% CI: 2.22, 3.54) compared to normotensive (**Table 2**).

## eGFR and CD4 cell count

The proportion with eGFR $<60$ml/min among those with a CD4 cell count $<100$ cells/uL was 15.5% (95% CI: 14.6, 16.6) compared to a 10.1% (95% CI: 9.9, 10.3%) among those with a CD4 cell count $>500$ cells/uL. Additionally, the univariate prevalence ratio among those with CD4 cell count $<100$ cells/uL was significantly different at 2.10 (95% CI: 1.88, 2.35) compared to those with a CD4 cell count $>500$ cells/uL (**Table 2**).

## eGFR and antiretroviral therapy regimen

Antiretroviral regimen data was available for 61,447 (89.6%) of which 58,214 (94.5%) received a TDF-containing first line combination, 2,964 (4.8%) received a non-TDF containing first line combination, and 205 ($<1\%$) received a protease inhibitor-containing regimen. Individuals prescribed a non-TDF-containing ART regimen had a significantly lower eGFR with a median of 78.4 compared to those on a TDF-containing ART regimen with a median of 99.8 (Pearson p-value: $<0.001$). Additionally, we illustrate the decreasing trend in proportion of individuals receiving a TDF-containing ART regimen with decreasing eGFR, with the exception of those with the lowest eGFR category ($<15$ ml/min/$1.73$m$^2$) (**Fig 2**).

## Changes in eGFR following ART initiation

There were 3,216 individuals in the analysis set with multiple creatinine measures spaced by a median of 210 days (IQR: 133–383 days). The distribution of eGFR measures show substantial heterogeneity across age groups and hypertensive status (**Fig 1A and 1B**). Among those with multiple eGFR measures the measures at 3–12 months of follow-up and $>12$ months tended to be higher than the ART initiation/baseline value (**Fig 3**).

**Table 1. Analysis population characteristics by CKD-EPI estimated glomerular filtration rate category at ART initiation visit.**

| Factor | Level | Normal | Mild | Moderate | Severe | Failure | p-value |
|---|---|---|---|---|---|---|---|
| N | | 43547 | 20257 | 3917 | 447 | 366 | |
| Sex | Female | 24906 (57.2%) | 13130 (64.8%) | 2444 (62.4%) | 246 (55.0%) | 213 (58.2%) | <0.001 |
| | Male | 18641 (42.8%) | 7127 (35.2%) | 1473 (37.6%) | 201 (45.0%) | 153 (41.8%) | |
| Age Category | <25 years | 7069 (16.2%) | 1328 (6.6%) | 179 (4.6%) | 30 (6.7%) | 32 (8.7%) | <0.001 |
| | 25–29 years | 9666 (22.2%) | 2709 (13.4%) | 376 (9.6%) | 53 (11.9%) | 59 (16.1%) | |
| | 30–34 years | 10480 (24.1%) | 4293 (21.2%) | 581 (14.8%) | 96 (21.5%) | 66 (18.0%) | |
| | 35–39 years | 8123 (18.7%) | 4183 (20.6%) | 736 (18.8%) | 70 (15.7%) | 76 (20.8%) | |
| | 40–44 years | 4436 (10.2%) | 3296 (16.3%) | 643 (16.4%) | 61 (13.6%) | 55 (15.0%) | |
| | 45–49 years | 2044 (4.7%) | 2008 (9.9%) | 473 (12.1%) | 50 (11.2%) | 36 (9.8%) | |
| | 50–54 years | 1031 (2.4%) | 1203 (5.9%) | 380 (9.7%) | 33 (7.4%) | 20 (5.5%) | |
| | 55–59 years | 415 (1.0%) | 651 (3.2%) | 245 (6.3%) | 25 (5.6%) | 13 (3.6%) | |
| | 60–64 years | 181 (0.4%) | 321 (1.6%) | 159 (4.1%) | 13 (2.9%) | 7 (1.9%) | |
| | 65 years | 102 (0.2%) | 265 (1.3%) | 145 (3.7%) | 16 (3.6%) | 2 (0.5%) | |
| Body Mass Index | Underweight | 1850 (4.2%) | 494 (2.4%) | 148 (3.8%) | 30 (6.7%) | 39 (10.7%) | <0.001 |
| | Normal Weight | 4487 (10.3%) | 1222 (6.0%) | 298 (7.6%) | 49 (11.0%) | 37 (10.1%) | |
| | Overweight | 891 (2.0%) | 277 (1.4%) | 72 (1.8%) | 11 (2.5%) | 6 (1.6%) | |
| | Obese | 293 (0.7%) | 126 (0.6%) | 32 (0.8%) | 3 (0.7%) | 5 (1.4%) | |
| | Unknown | 36026 (82.7%) | 18138 (89.5%) | 3367 (86.0%) | 354 (79.2%) | 279 (76.2%) | |
| Diabetes | Diabetic | 41 (0.1%) | 19 (0.1%) | 6 (0.2%) | 0 (0.0%) | 0 (0.0%) | 0.71 |
| | Missing | 43506 (99.9%) | 20238 (99.9%) | 3911 (99.8%) | 447 (100.0%) | 366 (100.0%) | |
| Blood Pressure Category | Hypotensive | 2424 (5.6%) | 881 (4.3%) | 243 (6.2%) | 47 (10.5%) | 40 (10.9%) | <0.001 |
| | Normotensive | 13677 (31.4%) | 5628 (27.8%) | 1115 (28.5%) | 128 (28.6%) | 97 (26.5%) | |
| | Pre-Hypertensive | 2397 (5.5%) | 1075 (5.3%) | 183 (4.7%) | 26 (5.8%) | 16 (4.4%) | |
| | Hypertensive Stage I | 4481 (10.3%) | 2228 (11.0%) | 429 (11.0%) | 50 (11.2%) | 38 (10.4%) | |
| | Hypertensive Stage II | 2277 (5.2%) | 1375 (6.8%) | 371 (9.5%) | 21 (4.7%) | 27 (7.4%) | |
| | Severe Hypertension | 121 (0.3%) | 126 (0.6%) | 49 (1.3%) | 5 (1.1%) | 1 (0.3%) | |
| | Unknown | 18170 (41.7%) | 8944 (44.2%) | 1527 (39.0%) | 170 (38.0%) | 147 (40.2%) | |
| ART Regimen | First Line Non-TDF | 1242 (2.9%) | 636 (3.1%) | 775 (19.8%) | 212 (47.4%) | 99 (27.9%) | <0.001 |
| | First Line TDF Containing | 37503 (86.1%) | 17407 (85.9%) | 2758 (70.4%) | 183 (40.9%) | 202 (55.2%) | |
| | PI Containing | 146 (0.3%) | 45 (0.2%) | 11 (0.3%) | 1 (0.2%) | 2 (0.5%) | |
| | Other First Line | 166 (0.4%) | 33 (0.2%) | 21 (0.5%) | 2 (0.4%) | 3 (0.8%) | |
| | Missing | 4490 (10.3%) | 2136 (10.5%) | 352 (9.0%) | 49 (11.0%) | 60 (16.4%) | |
| Year Care Initiated | 2011 | 8340 (19.2%) | 4404 (21.7%) | 819 (20.9%) | 95 (21.3%) | 63 (17.2%) | <0.001 |
| | 2012 | 8353 (19.2%) | 3752 (18.5%) | 686 (17.5%) | 83 (18.6%) | 58 (15.8%) | |
| | 2013 | 8158 (18.7%) | 4205 (20.8%) | 732 (18.7%) | 84 (18.8%) | 56 (15.3%) | |
| | 2014 | 7935 (18.2%) | 3925 (19.4%) | 675 (17.2%) | 82 (18.3%) | 54 (14.8%) | |
| | 2015 | 5560 (12.8%) | 2047 (10.1%) | 504 (12.9%) | 49 (11.0%) | 52 (14.2%) | |
| | 2016 | 4383 (10.1%) | 1648 (8.1%) | 423 (10.8%) | 46 (10.3%) | 60 (16.4%) | |
| | 2017 | 818 (1.9%) | 276 (1.4%) | 78 (2.0%) | 8 (1.8%) | 23 (6.3%) | |
| CD4 Cell Count | Median (IQR) | 296 (167–440) | 296 (166–434) | 236 (118–380) | 191 (81–333) | 221 (94–396) | <0.001 |
| CD4 Cell Count | >500 CD4 cell count | 5734 (13.2%) | 2562 (12.6%) | 367 (9.4%) | 26 (5.8%) | 36 (9.8%) | <0.001 |
| | 351–500 CD4 cell count | 7077 (16.3%) | 3544 (17.5%) | 478 (12.2%) | 39 (8.7%) | 33 (9.0%) | |
| | 251–350 CD4 cell count | 6456 (14.8%) | 3110 (15.4%) | 520 (13.3%) | 55 (12.3%) | 40 (10.9%) | |
| | 100–250 CD4 cell count | 8748 (20.1%) | 4234 (20.9%) | 915 (23.4%) | 95 (21.3%) | 70 (19.1%) | |
| | <100 CD4 Cell Count | 4530 (10.4%) | 2173 (10.7%) | 599 (15.3%) | 88 (19.7%) | 63 (17.2%) | |
| | Unknown | 11002 (25.3%) | 4634 (22.9%) | 1038 (26.5%) | 144 (32.2%) | 124 (33.9%) | |

Note: p-value for continuous variables were calculated with t-test and p-values for categorical variables were calculated with Chi-squared test, ART-antiretroviral therapy, IQR-interquartile range.

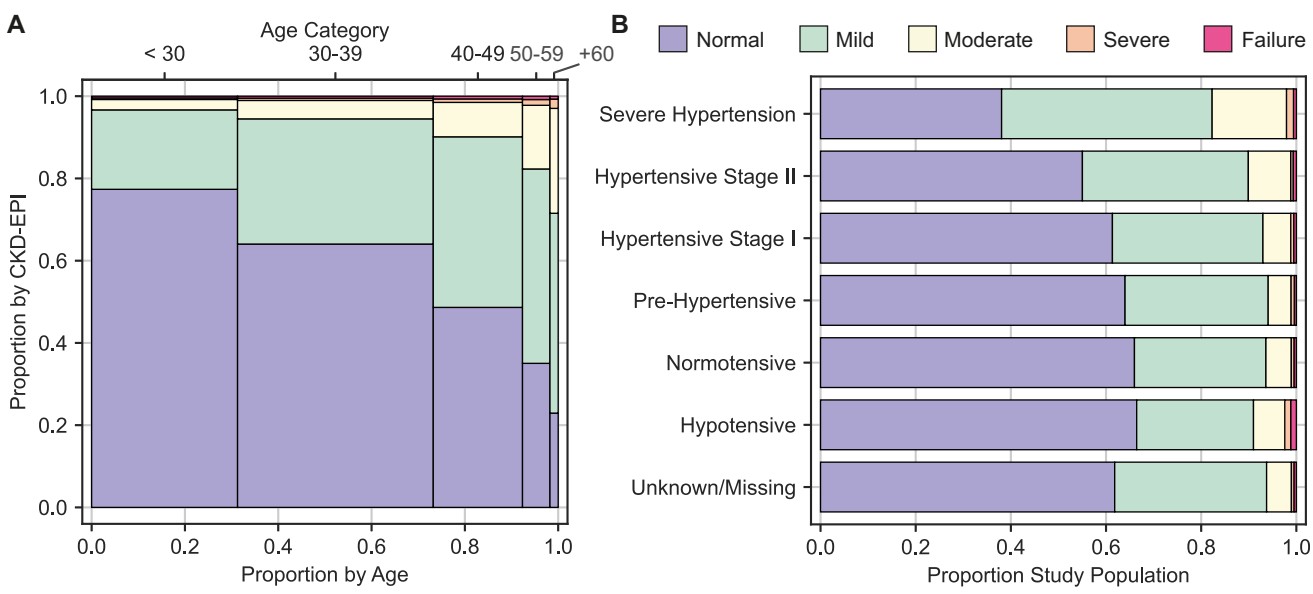

**Fig 1.** A) Mosaic Plot Estimated Glomerular Filtration Rate (eGFR) Category by Age Category B) Stacked Bar Chart eGFR Category by Hypertensive Category. Notes: The visit interval is 0 at ART initiation/baseline, individual follow-up time limited to two years.

## Adjusted model results

We calculated mixed effects Poisson regression estimates for eGFR <60ml/min (CKD-EPI) adjusted for age, sex, body mass index, blood pressure category, and CD4 cell count allowing for random effects at the facility level. There was a significant increase in adjusted prevalence of eGFR <60 ml/min associated with sex, age, blood pressure, and CD4 cell count (**Table 2**). Males were less likely to have a kidney function impairment with an adjusted prevalence ratio of 0.75 (95% CI: 0.70, 0.79) and those aged 65 years and older had the highest adjusted prevalence ratio at 10.39 (95% CI: 7.88, 13.70) compared to those aged 17–24 years. Blood pressure above 180mmHg systolic or above 120mmHg diastolic (severe hypertension) had the highest adjusted prevalence ratio of 1.64 (95% CI: 1.35, 1.98) followed by hypertension stage II with an adjusted prevalence ratio of 1.23 (95% CI: 1.11, 1.36) compared to normotensive. Low CD4 cell count defined as ≤100 cells/uL had the highest adjusted prevalence ratio of 1.79 (95% CI: 1.53, 2.09) compared to those with a CD4 cell count greater than 500 cells/uL (**Table 2**).

## eGFR across formulae

The Cockcroft-Gault equation resulted the lowest median eGFR at 90.3 ml/min/1.73m$^2$ (IQR: 73.7–110.4ml/min/1.73m$^2$) followed closely by the MDRD-4 equation at 90.7ml/min/1.73m$^2$ (IQR: 75.5–109.7ml/min/1.73m$^2$) and the Mayo Quadratic equation had highest median GFR estimates at 114.3ml/min/1.73m$^2$ (IQR: 103.6–123.1ml/min/1.73m$^2$). Adjustment for race in both the MDRD-4 and CKD-EPI equations significantly change the proportion of those categorized with mild and moderate kidney function impairment (**Fig 4** and **Table 3**). The race adjusted eGFR for moderate kidney function impairment by the MDRD-4 (7.4%, 95% CI: 7.2, 7.6) and CDK-EPI (5.6%, 95% CI: 5.5, 5.8) was significantly lower compared to the unadjusted eGFR for MDRD-4 (3.0, 95% CI: 2.9, 3.1%) and CKD-EPI (3.0, 95% CI: 2.9, 3.1%) (**Table 3**). Using the CKD-EPI formula as the comparator we found heterogeneity in kidney function impairment categorization with the four-variable modification of diet in renal disease

**Table 2. Crude and adjusted prevalence ratios for eGFR (CKD-EPI) <60ml/min/1.73m$^2$ at baseline.**

| Covariate | Level | Crude | | Adjusted | |
|---|---|---|---|---|---|
| | | PR | 95% CI | PR | 95% CI |
| Sex | Female | ref | ref | ref | ref |
| | Male | 0.95 | (0.90, 1.00) | 0.74 | (0.69, 0.79) |
| Age Category | 17–24 years | ref | ref | ref | ref |
| | 25–29 years | 1.36 | (1.17, 1.57) | 1.38 | (1.18, 1.62) |
| | 30–34 years | 1.71 | (1.48, 1.96) | 1.77 | (1.52, 2.04) |
| | 35–39 years | 2.37 | (2.07, 2.72) | 2.46 | (2.09, 2.89) |
| | 40–44 years | 3.20 | (2.79, 3.69) | 3.25 | (2.68, 3.93) |
| | 45–49 years | 4.33 | (3.75, 5.01) | 4.34 | (3.48, 5.40) |
| | 50–54 years | 5.83 | (5.02, 6.77) | 5.70 | (4.51, 7.20) |
| | 55–59 years | 7.64 | (6.51, 8.95) | 7.20 | (5.72, 9.07) |
| | 60–64 years | 9.54 | (8.00, 11.37) | 9.09 | (7.15, 11.55) |
| | 65+ years | 11.22 | (9.40, 13.40) | 10.18 | (7.83, 13.24) |
| Hypertension Category | Hypotensive | 1.43 | (1.27, 1.60) | 1.35 | (1.18, 1.53) |
| | Normotensive | ref | ref | ref | ref |
| | Pre-Hypertensive | 0.94 | (0.82, 1.07) | 0.92 | (0.81, 1.04) |
| | Hypertensive Stage I | 1.11 | (1.00, 1.22) | 1.03 | (0.94, 1.14) |
| | Hypertensive Stage II | 1.57 | (1.42, 1.74) | 1.23 | (1.11, 1.36) |
| | Severe Hypertension | 2.81 | (2.22, 3.54) | 1.63 | (1.34, 1.99) |
| | Unknown | 0.97 | (0.91, 1.04) | 1.04 | (0.95, 1.14) |
| CD4 Cell Count | >500 | ref | ref | ref | ref |
| | 351–500 | 1.00 | (0.89, 1.13) | 0.93 | (0.82, 1.05) |
| | 251–350 | 1.24 | (1.10, 1.40) | 1.11 | (0.97, 1.28) |
| | 100–250 | 1.62 | (1.45, 1.77) | 1.35 | (1.19, 1.52) |
| | <100 | 2.10 | (1.88, 2.35) | 1.72 | (1.48, 2.01) |
| | Unknown | 1.60 | (1.44, 1.77) | 1.43 | (1.25, 1.63) |

Note: PR–prevalence ratio, CI–confidence interval, crude and adjusted analysis allow random effect at the facility level.

with 86.3% agreement (Scott/Fleiss Pi: 0.75, 95% CI: 0.74, 0.75), followed by the Mayo Quadratic equation with 71.1% agreement (Scott/Fleiss Pi: 0. 26, 95% CI: 0. 25, 0.27) and Cockcroft-Gault with 69.4% agreement (Scott/Fleiss Pi: 0.44, 95% CI: 0. 44, 0.45). The MDRD-4 equation categorizes the majority (51.9%) with an eGFR <90ml/min/1.73m$^2$ while the Mayo categorizes the smallest proportion (11.89%) of individuals with an eGFR <90ml/min/1.73m$^2$ (**Fig 5**).

## Discussion

In this study we found that a substantial proportion (6.9%) of people with HIV initiating ART in Zambia have moderate to severely impaired kidney function. Furthermore, among those ≥50 years old a significant proportion of patients are experiencing kidney failure according to the CKD-EPI equation. We also provide evidence through the observed risk factors that the EMR may be used to aide identification of those with greater likelihood of a reduced eGFR.

The proportion of individuals with an eGFR <60ml/min was 1.7 times greater, though not significantly different, than previous estimates (4.1%, 95% CI: 3.3–7.1%) at 6.9% (95% CI: 6.7–7.1%) [13]. We also show a lower overall median eGFR (CKD-EPI) 99.1 (IQR: 81.4, 115.5) compared to previous work from Deckert et al in the Zambian setting with median eGFR

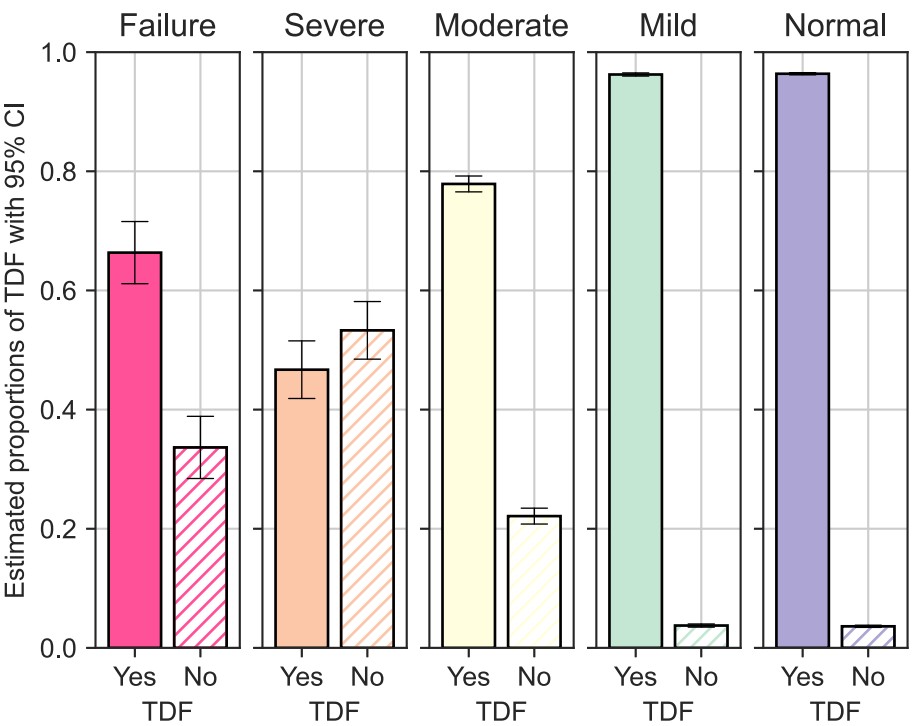

**Fig 2. Proportion TDF containing regimen by eGFR (CKD-EPI) category.**

(CKD-EPI) of 108.3 (IQR: 88.8, 118.5) as well as a larger proportion in the mild kidney function impairment category at 19.7% compared to 29.6% (95% CI: 29.2, 29.9% [13]). The median eGFR (per Cockcroft-Gault formula) of 90.5 ml/min/1.73m$^2$ (IQR: 73.7, 110.9) is 5.5ml/min/1.73m$^2$ lower than that reported by Mocroft et al at 96ml/min/1.73m$^2$ (IQR: 82, 111) [36]. Conversely, the proportion of those with severe kidney function impairment (CKD-EPI) presented here is lower at 0.6% compared to 0.9% [13]. As NCD research among the HIV population continues it will be important to track kidney function trends as ART regimens, diet, body mass index, blood pressure, and other population characteristics continue to change. Similarly, though it remains important to screen for TDF suitability and adjust ART regimens to reduce potential added renal stress, additional resources should be provided to monitor individuals receiving ART who are found to have impaired kidney function. We observed parallel decreasing in the proportion of individuals receiving a TDF contain regimen and eGFR category until eGFR<155ml/min/1.73m$^2$. It is possible that some individuals with low eGFR measures presented to the clinic as more ill and were initiated on a TDF-containing first line to avoid any delay in ART initiation and later switched to a non-TDF-containing regimen [37].

Zambia has made considerable progress regarding the UNAIDS 95-95-95 goals including decentralized clinics, differentiated services delivery model incorporation, and progressive HIV testing initiatives [38]. As Zambia continues advancement toward HIV epidemic control it is increasingly important to leverage laboratory measures, used for routine HIV care, to evaluate underlying non-communicable disease. These data serve as evidence that routine data may help jumpstart an understanding of the burden of kidney function impairment as well as

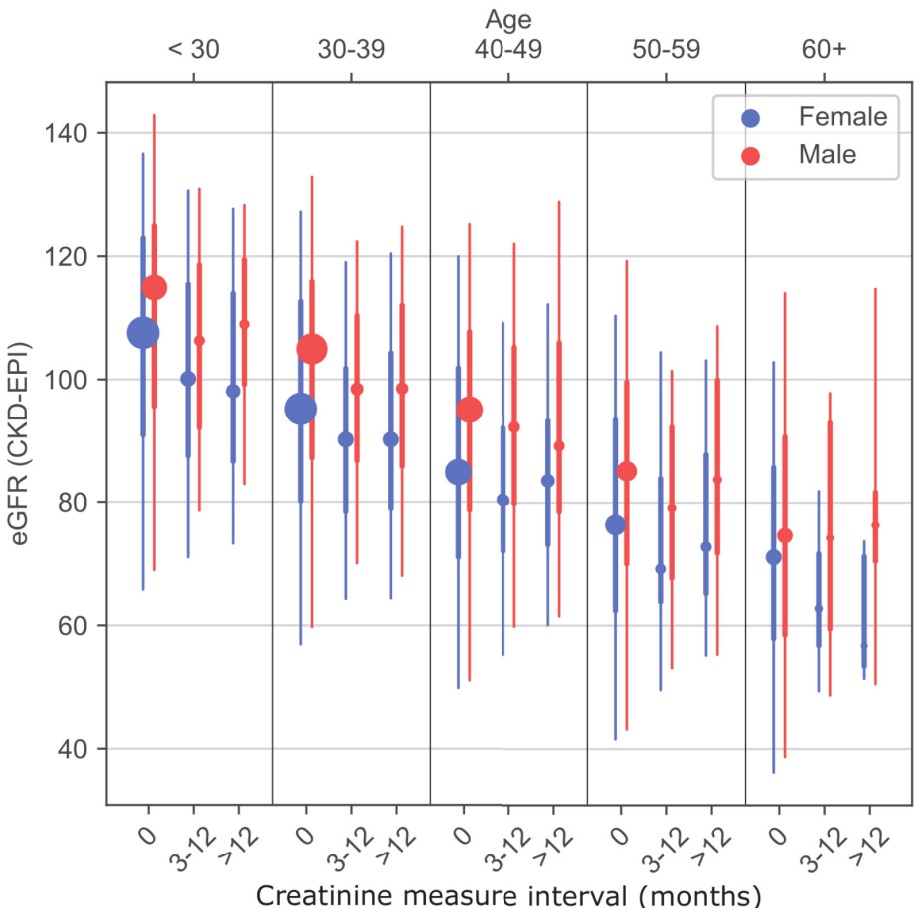

**Fig 3. Repeated estimated glomerular filtration rate measures by sex and age.** Notes: Marker size proportional to population size in each category.

guide the response to underlying non-communicable co-morbidities like kidney function impairment and high blood pressure.

During much of the study window (2011–2016) CD4 cell count was a part of routine ART initiation processes and remains an important indicator of HIV progression [22–24,39]. The association between kidney function and CD4 cell count aligns with previous work in Tanzania and Zambia [10,40]. It is possible that some observed kidney function impairment, especially severe kidney function impairment, could be related to HIV-associated nephropathy which was estimated to affect 33.5% of those with HIV in Zambia by Fabian et al in 2009 [41].

We calculated both adjusted and unadjusted eGFR using the CKD-EPI and MDRD-4 equations. The race adjusted proportion significantly lower of individuals in the normal, mild, and moderate categories compared to the not race adjusted for both the CKD-EPI and the MDRD-4 formulae. The proportion of those in the severe and kidney failure categories (eGFR < 30 ml/min/1.73m$^2$ for the race adjusted and not race adjusted CKD-EPI and MDRD-4 equations did not differ significantly. This could have implications for programs seeking to address kidney function impairment depending on the eGFR threshold for referral or renal care.

Previous research has found that Dolutegravir (DTG) may be associated with increased body mass index which may be an important upstream risk factor for kidney function

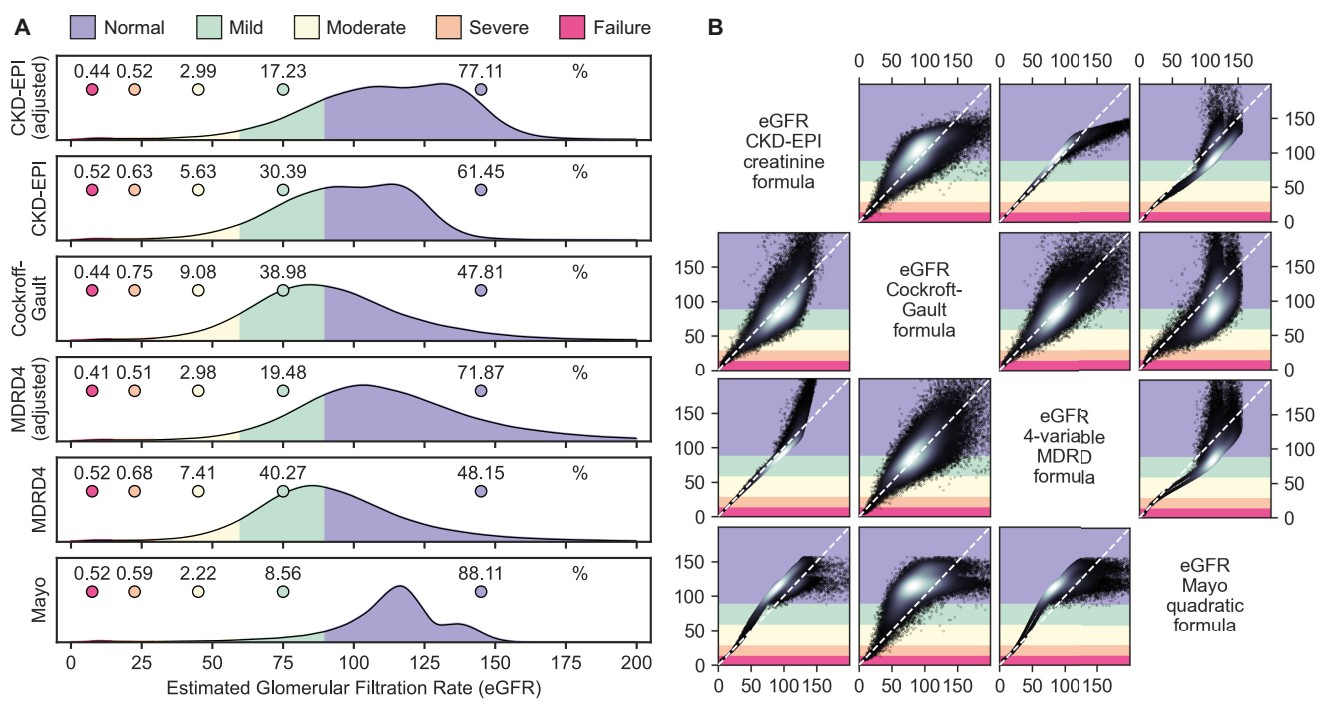

**Fig 4. Sankey diagram unadjusted and adjusted categorization for the CKD-EPI formula.**

impairment [42–44]. Though this analysis is limited to those on a non-DTG-based regimen it provides kidney function measures not confounded by the altered, often reduced, creatinine clearance that might contribute to kidney function misclassification among those on a DTG-containing regimen [45]. As Dolutegravir continues to be rolled out, it will be increasingly important to monitor potential DTG regimen associated risks for increased body mass index, as well as account for the more direct creatinine clearance effects among those receiving a DTG containing ART regimen.

There is a non-trivial amount of missing data in the EMR which may limit its utility as a tool for renal care referral and increased incorporation of routine measures into the Zambia HIV care guidelines. Zambia is in the process of implementing an electronic source documentation system, shifting away from the standard paper file registries, which, we anticipate will

**Table 3. Count of individuals estimated glomerular filtration rate (eGFR) category by eGFR formula.**

| eGFR Equation | eGFR Category | | | | | | | | | |
|---|---|---|---|---|---|---|---|---|---|---|
| | Normal | | Mild | | Moderate | | Severe | | Failure | |
| | Count | Percent (95% CI) | Count | Percent (95% CI) | Count | Percent (95% CI) | Count | Percent (95% CI) | Count | Percent (95% CI) |
| CKD-EPI | 45,824 | 62.8 (62.5, 63.2) | 22,163 | 30.4 (30.0, 30.7) | 4,107 | 5.6 (5.5, 5.8) | 462 | 0.6 (0.6, 0.7) | 377 | 0.5 (0.5, 0.6) |
| CKD-EPI Adj | 57,487 | 78.8 (78.5, 79.1) | 12,568 | 17.2 (17.0, 17.5) | 2,178 | 3.0 (2.9, 3.1) | 377 | 0.5 (0.5, 0.6) | 323 | 0.4 (0.4, 0.5) |
| MDRD-4 | 37,293 | 51.1 (50.8, 51.5) | 29,367 | 40.4 (39.9, 40.6) | 5,404 | 7.4 (7.2, 7.6) | 493 | 0.7 (0.6, 0.8) | 376 | 0.5 (0.5, 0.6) |
| MDRD-4 Adj | 55,880 | 76.6 (76.3, 76.9) | 14,209 | 19.5 (19.2, 19.8) | 2,170 | 3.0 (2.9, 3.1) | 372 | 0.5 (0.5, 0.6) | 302 | 0.4 (0.4, 0.5) |
| Mayo | 45,815 | 88.1 (87.9, 88.4) | 22,163 | 8.6 (8.4, 8.8) | 4,144 | 2.2 (2.1, 2.3) | 436 | 0.6 (0.5, 0.7) | 375 | 0.5 (0.5, 0.6) |
| Cockcroft-Gault | 34,066 | 50.8 (50.4, 51.1) | 26,169 | 39.0 (38.6, 39.4) | 6,098 | 9.1 (8.9, 9.3) | 503 | 0.8 (0.7, 0.8) | 294 | 0.4 (0.4, 0.5) |

Note: Percent calculated for equation/row; Adj—indicates adjustment for race.

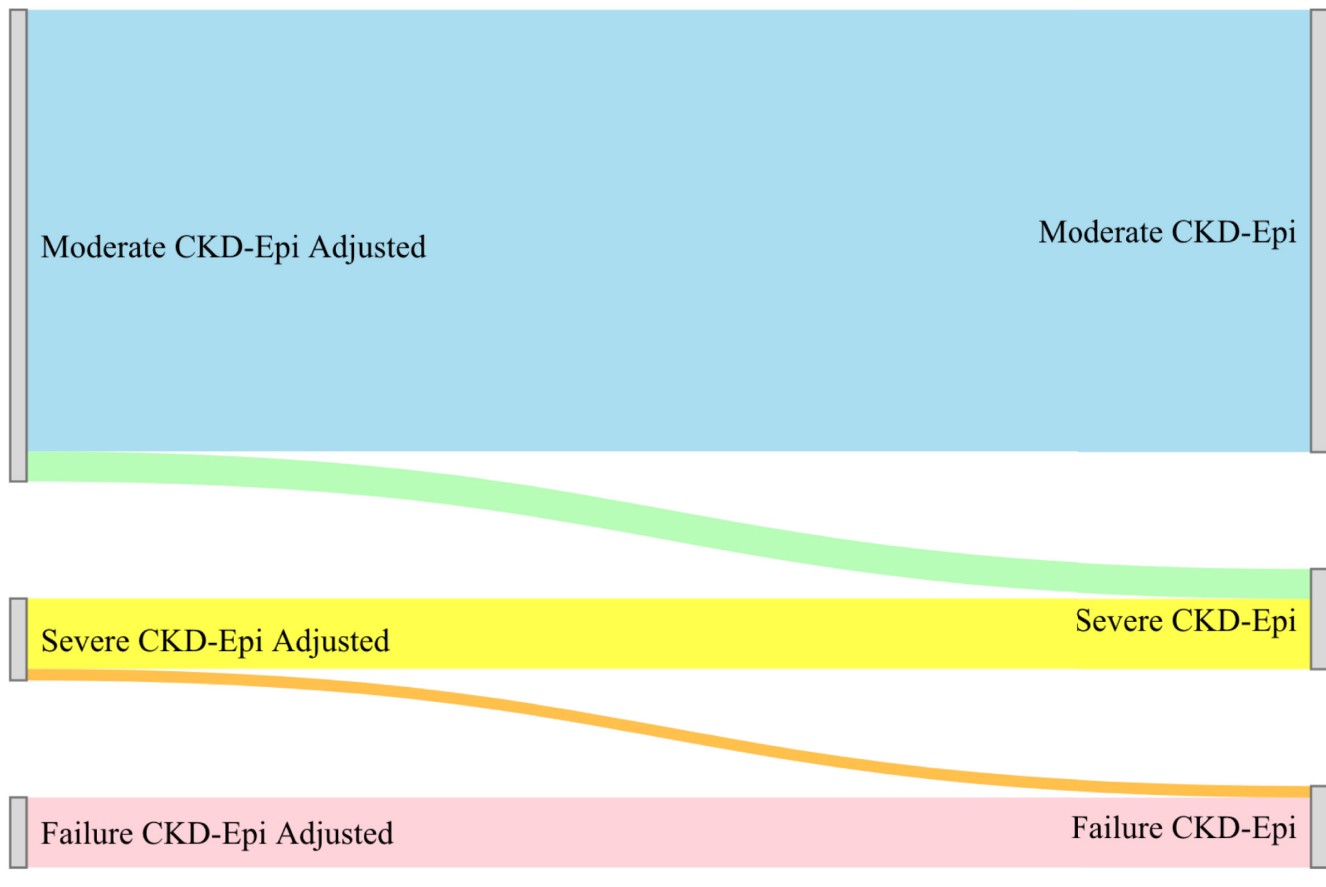

**Fig 5. Distribution of estimated glomerular filtration rates by formula.** Notes: Legend applies to Figs 4A and 4B. Category color corresponds to vertical axis in matrix. Deviation from the diagonal indicates disagreement between the two measures. White in scatter plot indicates density.

have a tremendous impact on data availability. This global push toward digital health records has already occurred in other parts of sub-Saharan Africa and not only helps bridge care geographically but allows clinicians to review analyzed data and bring to the fore potential underlying conditions [46–48].

Critical challenges to integrating NCD care at HIV care sites including limited clinic space, over-crowding, and availability of clinical staff to conduct screening and referral might be addressed through the continued uptake and expansion of differentiated HIV service delivery. It might be possible and important to design a differentiated service delivery model for those initiating HIV care with kidney function impairment [21,49,50].

This analysis represents one of the largest examinations of programmatic data on eGFR among those in HIV care in sub-Saharan Africa however, it does have limitations. One such limitation is the level of missing data fields in the electronic medical record. We explored trends in missingness comparing those in the larger parent dataset to those with a creatinine measure and found that missingness for CD4 cell count, body mass index (height and weight) was more common among those missing a creatinine measure however, we did not observe creatinine measure collection be restricted (e.g., advanced illness, older age group, body mass index category) to a subset of those seeking HIV care. The difference in CD4 cell count data is potentially associated with the year of HIV care initiation given that many creatinine measures were collected from 2011–2016, an era when CD4 cell count was part of the ART initiation

guidelines (S1 Table). Another limitation is limited medical notes and facility-level clinical context available to understand why some individuals had multiple creatinine measures in the medical record. We compared this subset of individuals in supplemental material (S2 and S3 Tables). Additionally, as we are not able to parse chronic kidney impairment from acute kidney injury (AKI) it is possible that some of those with decreased eGFR measures could be experiencing acute kidney injury and not necessarily indicative of chronic kidney disease. Despite these limitations, we are able, with the routine collection of data for HIV care to evaluate kidney function impairment including significantly associated contributors to decreased eGFR which represents an opportunity to use established infrastructure to address NCDs in Zambia among those initiating HIV care.

In conclusion, using routine serum creatinine measures, we identified a significant minority of PLHIV in Zambia initiating ART with moderate and severe kidney function impairment. Differentiated service delivery models could be a promising model to reinforce referral and kidney function monitoring among those initiating ART with kidney function impairment (eGFR $<60$ ml/min/1.73m$^2$).

## Supporting information

**S1 Fig. Scatter plot for eGFR (unadjusted CKD-EPI) and CD4 cell count (cells/mm$^3$) with linear fit line.**
(DOCX)

**S1 Table. Population characteristics by record of creatinine measure.**
(DOCX)

**S2 Table. Population characteristics by record of multiple creatinine measures.**
(DOCX)

**S3 Table. Crude and adjusted Prevalence Ratios (PR) for limited eGFR ($<$60ml/min/ 1.73m2).**
(DOCX)

## Acknowledgments

We would like to thank the Zambian Ministry of Health for making every effort to ensure those in HIV care continued to receive treatment. We would also like to thank the healthcare workers who faithfully delivered care to those receiving HIV care.

## Author Contributions

**Conceptualization:** Jake M. Pry, Michael J. Vinikoor, Carolyn Bolton Moore, Monika Roy, Harriet Daultrey, Andrew D. Kerkhoff, Elvin H. Geng, Brad H. Pollock, Jaime H. Vera.

**Data curation:** Jake M. Pry, Miquel Duran-Frigola, Jacob Mutale, Elvin H. Geng.

**Formal analysis:** Jake M. Pry, Miquel Duran-Frigola, Elvin H. Geng, Jaime H. Vera.

**Funding acquisition:** Elvin H. Geng, Jaime H. Vera.

**Investigation:** Jake M. Pry, Carolyn Bolton Moore, Aaloke Mody, Anjali Sharma, Elvin H. Geng, Jaime H. Vera.

**Methodology:** Jake M. Pry, Monika Roy, Aaloke Mody, Miquel Duran-Frigola, Andrew D. Kerkhoff, Elvin H. Geng, Brad H. Pollock, Jaime H. Vera.

**Project administration:** Jake M. Pry, Jaime H. Vera.

**Resources:** Jake M. Pry, Izukanji Sikazwe, Belinda Chihota, Andrew D. Kerkhoff.

**Software:** Jake M. Pry.

**Supervision:** Jake M. Pry, Anjali Sharma, Elvin H. Geng.

**Validation:** Jake M. Pry, Harriet Daultrey, Andrew D. Kerkhoff, Elvin H. Geng, Jaime H. Vera.

**Visualization:** Jake M. Pry, Miquel Duran-Frigola, Andrew D. Kerkhoff.

**Writing – original draft:** Jake M. Pry, Michael J. Vinikoor, Aaloke Mody, Anjali Sharma, Andrew D. Kerkhoff, Elvin H. Geng, Brad H. Pollock, Jaime H. Vera.

**Writing – review & editing:** Jake M. Pry.

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
