## [Decision Letter · Decision Letter 0]

23 Sep 2021

PGPH-D-21-00415

Evaluation of kidney function among people living with HIV initiating antiretroviral therapy in Zambia

Dear Dr. Pry,

Thank you for submitting your manuscript to PLOS Global Public Health. After careful consideration, we feel that it has merit but does not fully meet PLOS Global Public Health’s publication criteria as it currently stands. Therefore, we invite you to submit a revised version of the manuscript that addresses the points raised during the review process.

We look forward to receiving your revised manuscript.

Kind regards,

Siyan Yi, MD, MHSc, PhD

Academic Editor

Journal Requirements:

1. Please amend your detailed Financial Disclosure statement. This is published with the article, therefore should be completed in full sentences and contain the exact wording you wish to be published.

i). Please include all sources of funding (financial or material support) for your study. List the grants (with grant number) or organizations (with url) that supported your study, including funding received from your institution. 

ii). State the initials, alongside each funding source, of each author to receive each grant.

iii). State what role the funders took in the study. If the funders had no role in your study, please state: “The funders had no role in study design, data collection and analysis, decision to publish, or preparation of the manuscript.”

iv). If any authors received a salary from any of your funders, please state which authors and which funders.

2. Please ensure that the funders and grant numbers match between the Financial Disclosure field and the Funding Information tab in your submission form. Note that the funders must be provided in the same order in both places as well.

3. Please update the completed 'Competing Interests' statement, including any COIs declared by your co-authors. If you have no competing interests to declare, please state "The authors have declared that no competing interests exist". Otherwise please declare all competing interests beginning with the statement "I have read the journal's policy and the authors of this manuscript have the following competing interests:"

4. In the online submission form, you indicated that "De-identified data for this work will be made available upon request.". All PLOS journals now require all data underlying the findings described in their manuscript to be freely available to other researchers, either 1. In a public repository, 2. Within the manuscript itself, or 3. Uploaded as supplementary information.

5. Please provide separate figure files in .tif or .eps format only and remove any figures embedded in your manuscript file. Please ensure that all files are under our size limit of 20MB.  

Once you've converted your files to .tif or .eps, please also make sure that your figures meet our format requirements.

6. Tables should not be uploaded as individual files. Please remove these files and include the tables in your manuscript file.

7. We notice that your supplementary figures are included in the manuscript file. Please remove them and upload them  with the file type 'Supporting Information'. Please ensure that all Supporting Information files are included correctly and that each one has a legend listed in the manuscript after the references list. 

8. Please ensure that you refer to Figure 5 in your text as, if accepted, production will need this reference to link the reader to the figure.

9. Please match your Figures' Descriptions with their corresponding file names by renaming them accordingly.

Additional Editor Comments (if provided):

Reviewers' comments:

Reviewer's Responses to Questions

**Comments to the Author**

1. Does this manuscript meet PLOS Global Public Health’s publication criteria? Is the manuscript technically sound, and do the data support the conclusions? The manuscript must describe methodologically and ethically rigorous research with conclusions that are appropriately drawn based on the data presented.

Reviewer #1: Partly

Reviewer #2: No

2. Has the statistical analysis been performed appropriately and rigorously?

Reviewer #1: Yes

Reviewer #2: No

3. Have the authors made all data underlying the findings in their manuscript fully available (please refer to the Data Availability Statement at the start of the manuscript PDF file)?

Reviewer #1: No

Reviewer #2: No

4. Is the manuscript presented in an intelligible fashion and written in standard English?

Reviewer #1: Yes

Reviewer #2: No

5. Review Comments to the Author

Reviewer #1: The authors are to be congratulated on their manuscript entitled ‘Evaluation of kidney function among people living with HIV initiating antiretroviral therapy in Zambia’ which utilises a large national data set to address an important issue of renal disease in people living with HIV in Zambia. These data will add significantly to existing knowledge of the burden of renal impairment in PLWH and its risk factors and will contribute towards an emerging collection of much needed data for national and international non-communicable disease management policies.

There are some aspects which I think, if addressed, could improve the strength of the manuscript.

Major comments

Overall

1. Readers may benefit from a little more information on CIDRZ sites (perhaps there is a reference which can be included for this?). What is particularly important is giving an idea of the types of settings these data cover. Are they general outpatients, specific HIV clinics, inpatient settings, acute emergency care etc?

2. Further to this, it is critical that the reader has a clear understanding of who the included patients are from a clinical point of view. Is each data point one patient initiating ART (as indicated in the title)? Or are the data points from different stages across the HIV disease journey (as inferred in the abstract – “among a cohort of PLHIV with an HIV care visit” and the methods section – “All individuals with an HIV care clinic visit”)? If these are all data points from ART initiation and some are in an acute inpatient setting whilst others are in an outpatient clinic setting, it will be important to note that there may be significant differences within the cohort in terms of clinical disease. For example, those who have their HIV diagnosed as an inpatient may be much more clinically unwell, with sepsis for example, which may affect renal function.

3. The manuscript would benefit from standardisation of the terms used to define the outcome of renal impairment. These change throughout the manuscript which can make it difficult to follow (terms used include: eGFR measures, kidney impairment, moderate kidney function impairment, kidney function impairment, at least moderately impaired kidney function, at least moderately to severely impaired kidney function). Further, it is very important to differentiate between acute kidney injury and chronic kidney disease. It will be difficult to do this with data from one time point but this should be discussed, and implications addressed.

4. It would be nice to have an explanation of why the reported formulae were chosen. In particular, I would advise the authors to re-consider the inclusion of measures that adjust for race. There has been much international criticism of the inclusion of race in renal function calculations, arguing that race is a social construct rather than a biological one and including it in biological calculations disadvantages Black African populations.

Background

1. A little more detailed discussion of current knowledge of incidence and prevalence of AKI and CKD in PLWH in SSA would be beneficial

2. It might be useful to understand whether there are guidelines within the CIDRZ facilities for measurement of blood pressure, weight/height and diabetes, or whether these are carried out on an ad hoc basis when there is a clinical concern.

Methods

1. line 114. Why was there a modification to the AHA/ACC hypertension guidelines definition of severe hypertension?

Results

1. Overall, it is a little difficult to grasp the main findings of the study; this section might benefit from a little more focussing of the important messages

2. Line 214 “Obesity was also correlated….”. Please quote the numbers in the text to make this sentence more specific for the reader.

3. Line 215. “There was a substantial amount…”. Please report exactly how much data was missing (it is the vast majority of included participants).

4. Line 217. “.. the data available show a slightly higher proportion of…” It is unclear what is being compared here. Higher proportion than what?

5. Line 217. “Crude prevalence ratio…are very similar”. Again, unclear what is being compared here. Very similar to what? Also need to include the numbers in this sentence.

6. Overall, I would have reservations about reporting the obesity data at all given that it is in such a small proportion of the cohort and is likely to be subject to considerable bias (eg only being measured in those who are clinically unwell).

7. line 236. “There were 3,216 with multiple creatinine measures”. Please clarify 3,216 what. I presume PLWH included in the renal analysis cohort. Although this doesn’t seem to tie in with the earlier assertion that 3209 had multiple measurements?

8. Line 238: “Among those with at least…”. This sentence doesn’t make sense to me. I wonder if it would perhaps help to omit the part that says “Among those with at least two creatinine measures” as it seems to me that this analysis would have been done within the bigger cohort?

9. Line 241: “Among those with multiple eGFR measures at 3-12 months…”. It would be beneficial to have specific numbers here to support the observation. This finding would also make me concerned that (if the overall cohort was indeed from ART initiation timepoint), a lot of what has been observed is due to patients being clinically unwell at ART initiation, which then resolves with treatment and/or ART. This would lead to quite a different overall conclusion for the article if it were the case.

10. Line 247: It might be helpful to include a few words here on what the adjusted models are comparing

11. 248 -258: It's not clear from how this paragraph is phrased that it is reporting the results of an adjusted model. The first sentence could be along the lines of 'a logistic regression model examining cross sectional risk factors for moderate or severe renal impairment was constructed'. Independent risk factors included x,y and z (can list in order of association and give their odds ratios and confidence intervals after each).

Discussion

1. Line 282: “at least moderately to severely impaired kidney function” isn’t quite clear and doesn’t fit with definitions used in rest of text.

2. Line 296: “Our estimates for kidney function are also higher”. It is unclear what this sentence means.

3. Line 308: “chronic kidney disease as we show here”. I would have concerns about this. It is not clear to me that what is being presented in this paper is an assessment of chronic kidney disease (see comments above). This needs to be clarified.

Line 348: “We do not suspect differential bias to be associated with data missingness”. I would have concerns that there might indeed be bias with the risk factor data. Please provide information that would reassure the reader that these data are not biased (as per comment above on explaining local guidelines on weight measurement).

Line 349: “Another limitation is the ….”. This sentence is not clear to me. Is the measurement performed as standard of care, or is it routinely collected as clinical cause/judgement? In particular, is it not likely that those with repeated measurements are subject to clinical judgement?

Line 360: Again, point as above, I’m not sure the data presented in this paper has provided evidence on “those engaged in HIV care at increased risk for chronic kidney disease”.

Tables

Table 1

1. Looking at these data, I wonder what the p values are telling us practically. There are a lot of groups being compared. As a minimum, please insert a footnote with information on what statistical test was used and what it compared.

2. Conversely, there is an extremely high proportion of missingness for weight and diabetes categories and I wonder whether tests of statistical comparison are appropriate here?

3. I find it interesting that approximately half of patients in severe or failure categories are on TDF. This is despite the authors’ explanation that renal function is tested at ART initiation to help decide whether TDF can be given safely. I wonder whether this practice has changed throughout the course of the study. It might be worth a line of explanation on this.

Table 2

1. I find the linear correlation between CD4 T cell count and risk of renal impairment interesting. This could of course be related to acute kidney injury from intercurrent illness, but it might be worth highlighting this in the text rather than, or in addition to, comparing low category with high.

2. It would be beneficial to have a small footnote explaining what the analyses were adjusted for and how the models were constructed. This does not seem to be detailed in the methods section.

 

Minor comments

Abstract

Methods section should read “across seven of the ten” (instead of or)

Background

Line 89: ‘predictors of kidney impairment’. Can the authors please clarify that they assessed predictors in a longitudinal analysis, rather than cross sectional. If this is cross sectional analysis, can they please change to risk factors or associations?

Methods

line 126: “National Kidney Score”. Please indicate which nation this refers to.

Line 152: Please change to “Multiple imputation was considered where missingness was <30%”

Line 153: Please change to “Categories for BMI are defined according to World Health Organization criteria”.

Line 160: I’m not sure a description of what graphs were made is needed if you need to save words.

Results

Line 186: Please change to Prevalence of Kidney Function Impairment (or alternative standardised outcome term)

Line 190: “Severe impairment and kidney failure”. Perhaps “Severe impairment or kidney failure” might be clearer?

Line 225: Please specify if the protease inhibitor regime is also only first line as for TDF and non-TDF

Line 227: This sentence might read better if medians, IQR and p value were inserted together at the end.

Line 228: “Additionally, we illustrate….”. This sentence is unclear, I’m not sure what is being said here. Please rephrase.

Discussion

Line 285: I’m not sure what is referred to by “screen for TDF tolerance”. Perhaps “TDF suitability” might be a clearer term?

Line 296: There is a missing bracket at the end of the CI figures.

Line 308: I would delete the word “other” from this sentence as HIV is not a non-communicable disease

Line 318: please change effect to affect

Line 322: “As DTG is now a WHO recommended…” This sentence seems incomplete?

Line 329: change contain to containing

Line 332: not sure what is meant by “care referral and incorporation”?

Line 332: the word “is” is repeated

Line 340: “A critical challenge…”. Is this sentence complete? Refer for what?

Line 340: “It is possible with…” It isn’t clear what this sentence is trying to say.

Line 348: “fieldsin” requires a space

Line 351: this sentence is disjointed, please rephrase for clarity.

Acknowledgements

Line 365: You may wish to change received to receive.

Tables

Table 1: Please indicate whether they are all first line regimes or not.

Reviewer #2: 1. What is this manuscript all about?

In this study, they set out to determine the prevalence of kidney dysfunction/kidney failure defined as having moderate kidney dysfunction eGFR<60mL/min (moderate kidney function impairment) in unspecified/different censoring time within years. compare the different criteria for determining estimated glomerular filtration rate and model the predictors of the trajectory of kidney function following initiating of ART based therapy. They had a sample size of 68 534 with 72933 observations. They included anyone with at least baseline creatinine measurement and used mixed effect Poisson model to model moderate dysfunction which was defined as eGFR < 60ml/min/1.73m2.

General comment:

The research needs to be well focused with clearly outlined objectives to achieve. The analysis done and chose of statistical methods do not seem to meet the question they intended to answer and the conclusion made were not supported by data; this was true about the discussion as well. They had a lot of missing data and trivialized that fact in an interest to have a very large sample size. There is more analysis, review and probably effective methodological amendments they need to do to make the work clearer and publishable.

2. Have the authors identified the question and key claims and context in the introduction?

NO, this has not been well done. The research did not have a well-focused question to answer and seemed to be nebulous

3. Have they discussed related research? How does the study fit in the related research?

They have referenced some research but they have not tied in their study well and do not clearly demonstrate how theirs adds knew knowledge or innovation.

4. Do the figures and tables make sense given the results?

The tables may be combined for clarity. They also need a key for statistical methods used for the test of the null hypotheses and are better placed right below the results.

5. Methods and study design. Do the methods make sense and follow appropriate reporting guidelines?

The study design they mentioned was cross-sectional but it appears this was supposed to be a retrospective cohort study. They followed patients that for onset of moderate dysfunction after therapy.

6. Are the conclusions supported by the data and results?

NO, a lot needs to be done to make the manuscript up to standard for publication. They also need to do better in their discussion of the results and it should be done systematically from one result to another in a well-focused manner.

Figures/ tables are clearly presented and correctly labelled

Methods are detailed enough for another researcher can understand

Statistics are sound enough or further analysis is needed

Designs are appropriate for the question being asked or is there need for additional experiments

Are results supporting conclusions and are the data available?

References are missing and the title appropriate for the work done and informative.

Number comments and include page numbers.

MAJOR COMMENTS

They need to clarify how they estimate kidney failure in patients initiating treatment. Also explain if these patients were hospitalized and how they were followed up to determine Kidney Failure. They mentioned a number of endpoints and it made it unclear which one was their primary that they used for powering the study. There was kidney failure in the abstract, there was eGFR<60mL/min/1.73m2 and also moderate kidney dysfunction. How they defined kidney failure needed to come out clearly and at what time points they attempted to observe it.

They need to describe what they used as comparison group in this case and how long after the patient initiated therapy that they had their planned kidney function assessments.

They need to explain how many times those with repeated measurements had these measurements done to warrant the use of mixed effect Poisson methods and how many had more than two repeated measurements. Was there a specific follow up time? if not then others were followed after being treated for longer than others which increased their chances of dysfunction due to concomitant exposure to therapy. Also, there are chances of missing the outcome as the biomarkers stabilize after a long period from injury.

Since they included anyone with a baseline creatinine, and their inferences made on the entire population, they have to explain how they handled the missing follow up results for more than 90% (65000)of their participants.

With the very large sample size, where the observed difference clinically significant? A large sample size like this can show statistical significance that is not clinically significant. It was unclear why they opted for a complete enumeration when they had 3209 with repeated measurements from which they could have randomly sampled their study population and avoided all the missing data. Increasing the sample size may not lead to a different conclusion for a research but it may increase the precision.

There were many missing observations from line (236) of follow up visits more than 90% that was not explained how it was handled. They needlessly attempted to use a very large sample size that gave not extra new information. Mixed methods would be more effective for repeated measure usually more than 2 measures to model the trajectory of an outcome. Logistic regression, cox regression etc would have been better here. Proportional odds ordinal regression for the ordinal outcome on severity of kidney dysfunction.

They needed to focus their objectives; it appeared that they were interested in the predictors of eGR< 60; to finding the trajectory of the eGFR and method comparisons for eGFR formulae. These needed to be tied in well and focused. Respecting the method comparison, what was the reference method that gave the target eGFR?

They need to clarify their inclusion and exclusion criteria and justify that. e.g. did they include those with even those with previous kidney disorders? The enrollment process needs to be more elaborate.

They need a scientific or clinical basis for categorizing the variables such as age, BMI and blood pressure as they did. The arbitrary categorization which may not be linked to the clinical outcome are problematic.

They statistical analysis needs to be revisited or properly justified. They need to clarify whether the assumptions for using the parametric tests were met e.g. was eGFR normally distributed for t-test to be used? t-test, chi-square and mixed effects Poisson. They did not do any model diagnostics and validations to show the AUC, PPV, NPV, sensitivity and specificity of their model. Did not explain well in their methods how they came up with the predictors included in the model. They referred to univariate comparisons to make their inferences without adjusting for confounders.

Tables and figures had to follow. Tables showed be labeled on top and figures below. The information in the table showed be described right above. No key to show what statistical methods were used for the p-values. They did comparisons in the tables among predictors instead of outcomes. Then table 1 and 2 could be condensed into one table.

They mentioned diabetes as one of the covariates but there were no observations for this variable to include in the model (99.99 missing information.

MINOR COMMENTS

The subheadings for the results (line 174) can be put into one paragraph and the tables 1 and 2 can make one table that compares the different independent variables in relation to the outcome i.e. comparing independent variables among those who developed and those who did not develop the outcome. A well labelled table 1 with a key showing appropriate statistical methods used can be made.

A second table can be made from line 236 and address the change in eGFR. This can be to compare the baseline to after therapy and compare among the many dependent variables that can explain the change from baseline. Not comparing independent variables among themselves as in line 240.

It was not clear why the comparisons of the methods from line 263 to 273 were necessary. Why was this being done? These methods already have known differences and applications. They are not bound to give the same results in the first place

Lines 282-4 does not seem to be well backed by the evidence in the tables. It also did not show whether that was significant or whether that was from regression. The same is true about line 291 to 295.

Line 300 the Percents showed be presented with frequencies. e.g. 1/10 is (10%) and so is (100/1000).

The discussion from line 313 to 332 is not focused on the findings from the study or the data.

Recommendations in 341 and 442 are not supported by evidence from data in this paper

347 to 354 Missing data has a lot of chances to cause bias and wrong conclusions especially were the nature of the missingness is linked to the outcome or not by chance. With so much missing data it is not easy do rely on the findings and conclusions. And how do you use Poisson mixed effect models on cross sectional data observed just at baseline and no explanation of what happened to the missing observation and how they were addressed in the study.

. This is true for the entire conclusion section in lines 357 to 361.

6. PLOS authors have the option to publish the peer review history of their article (what does this mean?). If published, this will include your full peer review and any attached files.

**Do you want your identity to be public for this peer review?** For information about this choice, including consent withdrawal, please see our Privacy Policy.

Reviewer #1: **Yes: **Christine Kelly

Reviewer #2: **Yes: **FREEMAN W. CHABALA

---

## [Decision Letter · Decision Letter 1]

9 Jan 2022

Evaluation of kidney function among people living with HIV initiating antiretroviral therapy in Zambia

PGPH-D-21-00415R1

Dear Dr. Pry,

We're pleased to inform you that your manuscript has been judged scientifically suitable for publication and will be formally accepted for publication once it meets all outstanding technical requirements.

Within one week, you'll receive an e-mail detailing the required amendments. When these have been addressed, you'll receive a formal acceptance letter and your manuscript will be scheduled for publication.

An invoice for payment will follow shortly after the formal acceptance. To ensure an efficient process, please log into Editorial Manager at https://www.editorialmanager.com/pgph/ click the 'Update My Information' link at the top of the page, and double check that your user information is up-to-date. If you have any billing related questions, please contact our Author Billing department directly at authorbilling@plos.org.

Kind regards,

Siyan Yi, MD, MHSc, PhD

Academic Editor

Additional Editor Comments (optional):

Reviewers' comments:

Reviewer's Responses to Questions

**Comments to the Author**

1. If the authors have adequately addressed your comments raised in a previous round of review and you feel that this manuscript is now acceptable for publication, you may indicate that here to bypass the “Comments to the Author” section, enter your conflict of interest statement in the “Confidential to Editor” section, and submit your "Accept" recommendation.

Reviewer #1: All comments have been addressed

2. Does this manuscript meet PLOS Global Public Health’s publication criteria? Is the manuscript technically sound, and do the data support the conclusions? The manuscript must describe methodologically and ethically rigorous research with conclusions that are appropriately drawn based on the data presented.

Reviewer #1: Yes

3. Has the statistical analysis been performed appropriately and rigorously?

Reviewer #1: (No Response)

4. Have the authors made all data underlying the findings in their manuscript fully available (please refer to the Data Availability Statement at the start of the manuscript PDF file)?

Reviewer #1: Yes

5. Is the manuscript presented in an intelligible fashion and written in standard English?

Reviewer #1: Yes

6. Review Comments to the Author

Reviewer #1: All comments have been addressed and the manuscript now reads clearly.

7. PLOS authors have the option to publish the peer review history of their article (what does this mean?). If published, this will include your full peer review and any attached files.

**Do you want your identity to be public for this peer review?** For information about this choice, including consent withdrawal, please see our Privacy Policy.

Reviewer #1: **Yes: **Dr Christine Kelly
